# Sound Absorption Improvement in Porous Ferroelectret Polyethylene with Effective Piezoelectric Mechanism

**DOI:** 10.3390/polym14224843

**Published:** 2022-11-10

**Authors:** Yasmin Mohamed Yousry, Eleftherios Christos Statharas, Kui Yao, Ayman Mahmoud Mohamed, Poh Chong Lim, Francis Eng Hock Tay

**Affiliations:** 1Institute of Materials Research and Engineering, A*STAR (Agency for Science, Technology and Research), 2 Fusionopolis Way, Innovis, Singapore 138634, Singapore; 2Department of Mechanical Engineering, National University of Singapore, 9 Engineering Drive 1, Singapore 117575, Singapore

**Keywords:** acoustic absorbers, ferroelectret, piezoelectric, polyethylene, foam, noise mitigation

## Abstract

Airborne sound absorption in porous materials involves complex mechanisms of converting mechanical acoustic energy into heat. In this work, the effective piezoelectric properties of polyethylene ferroelectret foams on sound absorption were investigated by comparable samples with and without the piezoelectric response. Corona poling and thermal annealing treatments were applied to the samples in order to enable and remove the piezoelectric property, respectively, while the microstructure and the mechanical properties remained substantially unchanged. The effective piezoelectric properties and airborne sound absorption coefficients of the polyethylene foam samples before and after material treatments were measured and analyzed. Our experimental results and theoretical analysis showed that the open-cell ferroelectret polymer foam with an effective piezoelectric property provides an additional electromechanical energy conversion mechanism to enhance the airborne acoustic absorption performance.

## 1. Introduction

Noise pollution is an environmental issue that significantly affects the health and life quality, particularly for people living in high-density urban areas [1]. Various sources of noise in cities, from construction to land traffic and airplane noise, create harmful and unpleasant living and working environments. The methods for noise mitigation can typically be grouped into three major categories: (i) passive, (ii) active, and (iii) hybrid [2,3,4]. As the most extensively used method effective for mid- and high-frequency airborne noise mitigation, the passive approach can absorb noise by passively converting acoustical energy into heat dissipation, in which the working mechanisms include viscous effect, thermal effect, and material damping [5,6,7,8,9]. The viscous effect is the cause of the adherence of the fluid in the interface with the solid. The thermal effect is due to the dynamic heat conduction forced by the rarefaction and densification of the air. Material damping offers another form of mechanical energy dissipation when the sound wave impacts on the solid [10,11,12,13].

The active noise control method, which reduces unwanted noise by superimposing an antiphase signal of the same frequency through electric control circuitry, is more effective in lower frequency ranges [3]. Passive systems with inner resonators could also be tailored for lower-frequency noise mitigation [14]. Hybrid noise control deploys both active noise control for low-frequency regions and passive control for mid- and high-frequency range. Piezoelectric technologies are typically used as sensors or actuators in the active noise control method [15,16] and in shunt damping circuits in the hybrid method [17,18]. Piezoelectric materials are also used as passive dampers to reduce structure-borne vibration and sound radiation from a structure [19]. There are very limited studies which have explored the piezoelectric property in effective passive airborne noise absorption. In our previous works, we demonstrated superior airborne sound absorption performance from open-cell semi-conductive polar polyvinylidene fluoride (PVDF) foams, which is attributed to hybrid local piezoelectric and conductive functions [20,21,22,23,24]. In addition to piezoelectric polymers with intrinsic piezoelectric properties, ferroelectrets are cellular polymers that exhibit ferroelectric-like behavior after their exposure to high electric fields, and they have wide applications in producing pressure sensors, ultrasound transducers, microphones, and speakers. Ferroelectret cellular polymers could be cheaper and more widely available than ferroelectric polymers. This motivates us to demonstrate enhanced airborne sound absorption mechanism and performance with a ferroelectret, i.e., introducing and utilizing the extrinsic but effective piezoelectric effect in nonpolar polymer foams through electrical charging. In this work, we aim to explore the airborne noise absorption effect of porous ferroelectret made of non-polar polyethylene (PE) and further clarify the contribution of effective piezoelectric property on airborne noise absorption, through both experiment and theoretical analysis.

## 2. Materials and Methods

To determine the contribution of the piezoelectric effect on noise mitigation, we needed to obtain samples with comparable morphologies and structures, but with and without the piezoelectric property. For this purpose, non-polar PE foams from Sealed Air Pte. Ltd. (Charlotte, NC, USA) were first cut into circular discs with dimensions of 3 mm thicknesses and 29 mm diameters. Then, the foam samples were activated as ferroelectret with effective piezoelectric property after electrical charging by corona poling treatment. A high-voltage source was connected to the corona needle, and the PE foam was placed on a stainless-steel substrate connected to the ground, as shown in Appendix A (from the Appendix A), for a photograph of the corona poling setup. When the corona discharge was triggered by the application of a high voltage, electric charge from the corona needle was sprayed onto the top surface of the PE foam, creating a poling electric field between the top and bottom surface of the PF foam. Poling voltages of 45 kV and 55 kV were chosen to be high enough to activate the ferroelectret property in the PE foams, which were placed 20 mm away from the corona needle and treated for 3 min. The space charges in the foam were expected to stay trapped for an extended period of time [25]. The ferroelectret samples were subsequently annealed at 80 °C for 1 h in order to remove the space charges and the piezoelectric property [26,27].

The sound absorption property of the PE foams was tested using the impedance tube method based on the ISO 10534–1:1996 standard before and after the corona poling process, and finally after discharge by annealing treatment.

## 3. Results

The results regarding the effects of corona poling and annealing on the microstructure, dielectric, and mechanical properties of the foams are provided in the Appendix A.

### 3.1. Sound Absorption Measurements

Figure 1a presents the sound absorption coefficient of the PE foams with thicknesses of only 3 mm in the frequency range of 500–6400 Hz before and after corona poling. Figure 1b shows the sound absorption coefficient in the same frequency range after annealing at 80°C for 1 h. The measurement results in Figure 1a clearly indicate that the sound absorption coefficient of the PE foam samples increased after corona poling with voltages of 45 kV and 55 kV, while the results in Figure 1b show that the sound absorption coefficient of all the samples changed to approximately the original value after the space charge was removed by the annealing process.

### 3.2. Analyses and Discussion

To understand the mechanism underlying the observed changes in the sound absorption property of the PE foams, the effects of corona poling and annealing on the morphology, mechanical, dielectric, and piezoelectric properties of the PE foam samples were examined. Appendix A, presents the morphology of PE foam samples at the same locations before and after corona poling at 55 kV for 3 min, showing no substantial effect of corona poling on the morphology of the PE foams. Appendix A, present the measured stiffness of PE foam samples with and without corona poling before and after annealing treatment at 80 °C for 1 h. The results show very minor effects on the stiffness of the foams from the corona poling or the annealing treatments.

Furthermore, the tan delta of the foam, which is a property that shows the mechanical energy loss of the foam and is calculated from the ratio of loss modulus to the storage modulus, is measured and presented in Appendix A, for the PE foams with and without corona poling before and after annealing. The results show very minor effects on the tan delta of the foams from the corona poling and the annealing. In contrast, corona poling and annealing treatments significantly changed the dielectric and piezoelectric properties of the PE foams. This is shown in Appendix A, where the dielectric constant of 2.5 of the original PE foam sample increased to 3.2 after poling with 45 kV and 3.7 with 55 kV. The dielectric constant returned to almost the same value of 2.5 after annealing at 80 °C, as shown in Appendix A.

The piezoelectric properties of the foam samples were tested using a mechanical shaker (Tira TV 51110C, TIRA GmbH, Schalkau, Germany). The foam sample was sandwiched between two Cu tapes as top and bottom electrodes. The sample was placed on the top of a shaker, and a mass of 85 g was placed on top of it. An amplified sinusoidal driving signal from a function generator was sent to the vibrating shaker through a power amplifier. The output voltage generated by the sample in response to the vibration from the shaker was measured by an oscilloscope to evaluate the piezoelectric response. The shaker excited the sample at a frequency of 100 Hz and acceleration of 1 m/s^2^. The piezoelectric strain coefficient *d_33_* was calculated using Equation (1) [28]:(1)$$d_{33}=\frac{V\varepsilon_{33}A}{Mah}$$
where *V* is the output voltage, *ε_33_* is the dielectric constant, *A* is the cross-sectional area across which the force is applied, *M* is the mass of the applied load, *a* is the applied acceleration, and *h* is the thickness of the foam sample. The output voltage from the PE foam samples poled at 45 kV and 55 kV is presented in Figure 2 compared to the foam sample without the corona poling treatment. The output voltage from the original sample without corona poling was 12 mV, and it increased rapidly to 210 mV for the sample poled at 45 kV and 240 mV for the sample poled at 55 kV. The calculated piezoelectric coefficients for the PE foam samples without and with poling and after annealing are given in Table 1. The results in Table 1 show that the PE foam did not have a substantial piezoelectric effect before the corona poling, but it became highly piezoelectric active as a ferroelectret after the corona poling treatment. Increasing the corona poling voltage further improved piezoelectric *d_33_*, reaching 21 pC/N, corresponding to a poling voltage of 55 kV. However, after the annealing process at 80 °C, the piezoelectric *d_33_* coefficient almost vanished completely.

The experimental results above show that corona poling enhanced the sound absorption performance of PE foam, and the enhancement became more significant in the higher frequency range, particularly above 1500 Hz. The increase in sound absorption could be possibly attributed to the piezoelectric effect, both of which were simultaneously enabled with the corona process and eliminated after annealing. The more frequent vibrations of the ferroelectret foam induced by higher-frequency sound could facilitate mechanical to electrical energy transduction through the piezoelectric effect. To verify the contributions from the piezoelectric effect, we analyzed the acoustic power flowing through the foam and the power absorbed by the foam before and after corona poling, in comparison with the electric power generated from the foam. 

The Input acoustic power $P_{in}$ to the foam sample can be calculated according to Equation (2) [29]:(2)$$P_{in}=\frac{Ap^{2}}{\rho c}\cos\theta$$
where *A* is the surface area of the sample (6.6 cm^2^), *p* is sound pressure which can be obtained from Equation (3) [29], *ρ* is the air density (1.225 kg/m^3^), *c* is speed of sound in air (343.2 m/s), and *θ* is the angle between the direction of the sound propagation and the normal to the surface (zero degrees in our case).
(3)p=pref 10SPLdB20
where *P_ref_* is the reference sound pressure with a constant value of 20 µPa, and *SPL* is the sound pressure level which was maintained at 85 dB during the measurement. The calculated input acoustic power is 70.53 nW. The absorbed acoustic power is (0.45 × 70.53 nW), (0.50 × 70.53 nW), and (0.52 × 70.53 nW) for the samples poled at 0, 45, and 55 kV, respectively.

In order to determine the output electrical power generated from the foam samples, their voltage outputs in response to the acoustic excitation were measured in the impedance tube. After the PE foam samples were subjected to corona poling, silver paste was deposited on each side of the samples as the electrodes. The samples were positioned in the impedance tube, and the voltage outputs were measured with an oscilloscope connected to the electrodes during an acoustic excitation of 85 dB at different frequencies. The electrical power generated at the sample is given by Equation (4):(4)$$P_{out}=\frac{V_{out}^{2}}{2Z}$$
where *V_out_* is the voltage output and *Z* is the electrical impedance of the sample. Figure 3a shows the voltage outputs for the two foam samples poled at different voltage in the acoustic excitation at 6400 Hz, while Figure 3b shows the output electric power generated by the samples over the frequency spectrum, as well as the difference of the acoustic power absorbed by the two samples before and after corona poling at 45 kV and 55 kV.

The results in Figure 3b show that the electric power outputs from the three samples follow the same trend with the increased acoustic power absorbed after corona poling at the higher voltage of 55 kV, with the dependence on frequency and poling voltage. A detailed presentation of the analysis on the data at 6400 Hz is provided In Table 2, including the values of the additional acoustic power absorbed after poling of the PE foam samples at 45 kV and 55 kV in comparison with the electrical power generated. The results show that the electrical power generated from the samples (2.1 nW and 1.1 nW for the samples poled at 55 kV and 45 kV, respectively) was in the same order of magnitude as the additional acoustic power absorbed after corona poling (5.0 nW and 3.5 nW, respectively).

### 3.3. Theoretical Model

The above results imply that the effective piezoelectric effect, enabled by electrical poling, substantially contributes to enhanced noise mitigation. As observed in Figure 3b and Table 2, there is a difference between the generated electrical power and the difference in the absorbed acoustic power. The difference is probably due to the energy losses that happened during the mechanical-to-electrical energy conversion and electrical energy collection process, and even any other possible energy dissipation effects. For example, the charged surfaces of the pores after poling may also enhance the friction loss, which in turn improves the sound absorption performance of PE foams but without contributing to the observed electrical output. Figure 4 provides a schematic illustration of the mechanical energy dissipation mechanisms in ferroelectret PE foams before and after poling. When the airborne sound wave strikes an open-cell porous material, the sound pressure wave penetrates into the material, causing the energy transfer from moving air to the solid structure. Besides the conventional viscous, thermal, and material damping effects, the PE foam after poling has at least additional two effects: (a) mechanical vibration excited by the sound wave is partially damped by converted to space charges through the effective piezoelectric mechanism and finally to heat, and (b) material damping at the air–solid interface is enhanced in the presence of space charges. Our experimental results and theoretical analysis show the effectiveness of electric poling that enables ferroelectret PE foams to have great potential for noise mitigation applications.

## 4. Conclusions

The effective piezoelectric property of polyethylene ferroelectret foams on airborne sound absorption was investigated with dedicatedly designed samples comparable between foams with and without the piezoelectric response. The samples were selectively treated by corona poling and a controlled thermal annealing process to enable and remove the piezoelectric property, respectively, without substantially changing the samples’ structure and mechanical property. The measurement results showed that the sound absorption performance of the polyethylene foams improved with the presence of space charges and effective piezoelectric effect. The further analyses on the experimental results indicated that the piezoelectric property of the ferroelectret foams provided an additional sound absorption mechanism by converting the mechanical energy into electricity. Our experimental results and theoretical analysis suggest that effective piezoelectric response can be introduced in porous materials to enhance their airborne acoustic absorption.

## Figures and Tables

**Figure 1 polymers-14-04843-f001:**
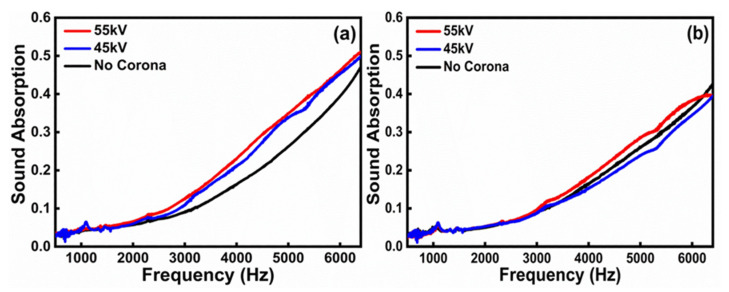
Sound absorption coefficient for PE foams with thicknesses of only 3 mm in the frequency range of 500–6400 Hz, (**a**) before and after corona poling at different voltages, and (**b**) after subsequent annealing treatment. Note the small but well-distinguishable change is due to the thickness constraint of only 3 mm, in contrast to standard sample thickness of 25 mm. The small thickness was selected to realize effective corona poling effect for the mechanism study. To realize a significant level of sound absorption for most practical applications, multilayers of small-thickness foams can be used.

**Figure 2 polymers-14-04843-f002:**
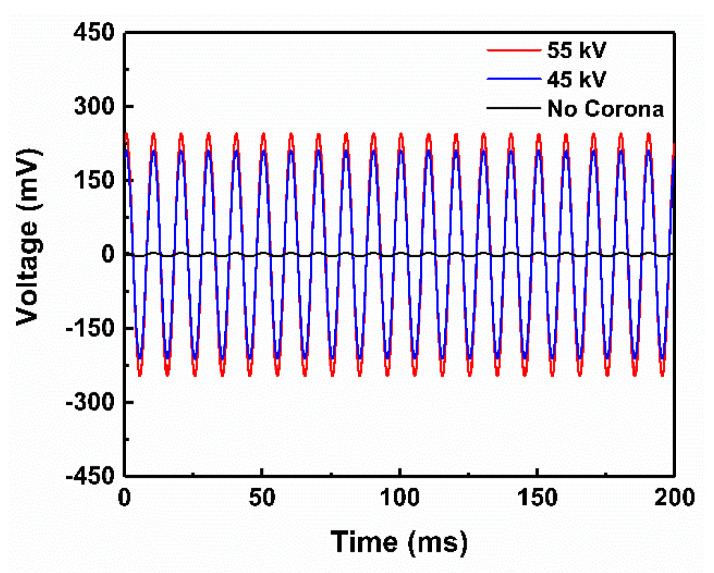
Voltage measurement output from a foam sample poled at 0, 45, and 55 kV and excited by shaker.

**Figure 3 polymers-14-04843-f003:**
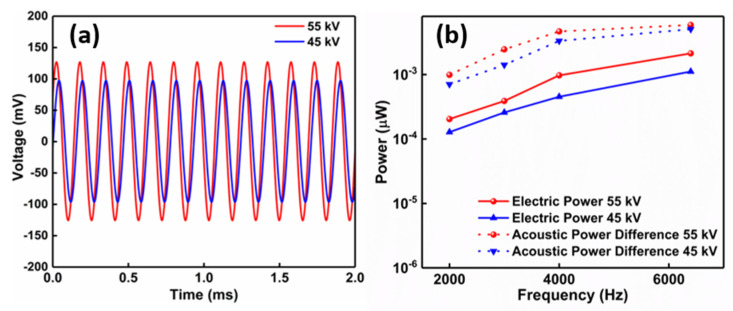
(**a**) Voltage outputs from the PE foam samples poled at different poling voltages under excitation of 85 dB at 6400 Hz, and (**b**) electric power outputs and difference of acoustic power in the two samples poled at different poling voltages under excitation of 85 dB at various frequency values.

**Figure 4 polymers-14-04843-f004:**
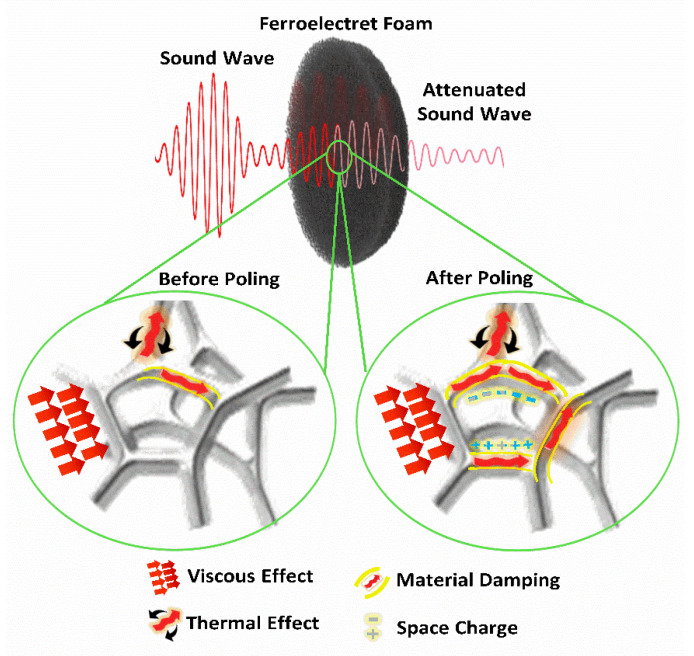
A schematic illustration of the mechanical energy dissipation mechanisms in ferroelectret foam with the presence of space charges and additional material damping after corona poling in comparison with before the poling.

**Table 1 polymers-14-04843-t001:** Piezoelectric coefficients of the foam samples without and with poling, before and after annealing.

Corona Voltage (kV)	*d_33_* After Corona (pC/N)	*d_33_* After Annealing (pC/N)
0	<1	<1
45	17	<1
55	21	<1

**Table 2 polymers-14-04843-t002:** Additional acoustic power absorbed after corona poling and electrical power generated for the PE foam samples (6400 Hz).

Parameter	Value		
Poling voltage (kV)	0	45	55
Input acoustic power (nW)	70.5	70.5	70.5
Total acoustic power absorbed (nW)	31.7	35.2	36.7
Output voltage (mV)	0	100	130
Additional acoustic power absorbed after poling (nW)	–	3.5	5.0
Experimental electrical power generated (nW)	–	1.1	2.1

## Data Availability

Not applicable.

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
