# Peer review of "Sound Absorption Improvement in Porous Ferroelectret Polyethylene with Effective Piezoelectric Mechanism"

_polymers, 2022, doi:10.3390/polym14224843_

Round 1

Reviewer 1 Report

The paper presents an interesting work on the effective piezoelectric properties of polyethylene ferroelectret foams for sound absorption. However, some essential results are missing, and necessary discussion is insufficient. I do not recommend publishing this work without addressing the following issues:

(1) In general, PE is not beneficial to piezoelectric effect. Why did the authors select PE to prepare the foams?

(2) The sound absorption coefficients of the PE foams with the thickness of 3 mm dont reach a significant level for practical application. In this view, it is inevitable to increase the thickness of the PE foams. Dose the polarization methodology described in this article work for thicker foams?

(3) The measurement details of the piezoelectric properties should be provided.

(4) The results show that the piezoelectric property of the PE foams is beneficial to improve the sound absorption property at high frequency. However, the improvement is limited by the thickness and it is difficult to anticipate the sound absorption property of the thicker PE foams.

(5) Moreover, it is pivotal to improve the sound absorption property at low frequency rather than at high frequency, because the sound absorption coefficients can exceed 0.95 at high frequency in many previous works. However, the results in this work didnt show any advantage of the PE foams in sound absorption at low frequency.

Reviewer 2 Report

This is an interesting manuscript reporting the sound absorption of porous ferroelectret PE mediated by effective piezoelectric mechanism. All the experiments were reasonably designed, and all the data were presented in satisfactory details. The manuscript was well-written. It is acceptable for publication with minor revision.

(1) In the revision the authors should describe the corona poling method in detail. 

(2) What is the mechanism of the enhanced dielectric permittivity after corona poling?

Round 2

Reviewer 1 Report

This manuscript reports on the effective piezoelectric properties of polyethylene ferroelectret foams for sound absorption. The results and analyses are both interesting and valuable, and the previous deficiencies and errors have been modified. This work could be considered for publication.